# Prediction of risk of prolonged post-concussion symptoms: Derivation and validation of the TRICORDRR (Toronto Rehabilitation Institute Concussion Outcome Determination and Rehab Recommendations) score

Laura Kathleen Langer[1]*, Seyed Mohammad Alavinia[1,2], David Wyndham Lawrence[3,4,5], Sarah Elizabeth Patricia Munce[1,6,7], Alice Kam[3], Alan Tam[2,3], Lesley Ruttan[3,8], Paul Comper[3,4,7], Mark Theodore Bayley[1,2,3]

1 KITE Research Institute at Toronto Rehabilitation Institute–University Health Network, Toronto, Canada, 2 Faculty of Medicine, Division of Physical Medicine and Rehab, University of Toronto, Toronto, Canada, 3 Toronto Rehabilitation Institute–University Health Network, Toronto, Canada, 4 Faculty of Kinesiology and Physical Education, University of Toronto, Toronto, Canada, 5 Mt Sinai Hospital, Toronto, Canada, 6 Department of Occupational Science & Occupational Therapy, University of Toronto, Toronto, Canada, 7 Rehabilitation Sciences Institute and Institute of Health Policy, Management & Evaluation, University of Toronto, Toronto, Canada, 8 Graduate Department of Psychological Clinical Science, University of Toronto Scarborough, Toronto, Canada

* laura.langer@uhn.ca

## Abstract

### Background

Approximately 10% to 20% of people with concussion experience prolonged post-concussion symptoms (PPCS). There is limited information identifying risk factors for PPCS in adult populations. This study aimed to derive a risk score for PPCS by determining which demographic factors, premorbid health conditions, and healthcare utilization patterns are associated with need for prolonged concussion care among a large cohort of adults with concussion.

### Methods and findings

Data from a cohort study (Ontario Concussion Cohort study, 2008 to 2016; $n = 1,330,336$) including all adults with a concussion diagnosis by either primary care physician (ICD-9 code 850) or in emergency department (ICD-10 code S06) and 2 years of healthcare tracking postinjury (2008 to 2014, $n = 587,057$) were used in a retrospective analysis. Approximately 42.4% of the cohort was female, and adults between 18 and 30 years was the largest age group (31.0%). PPCS was defined as 2 or more specialist visits for concussion-related symptoms more than 6 months after injury index date. Approximately 13% (73,122) of the cohort had PPCS. Total cohort was divided into Derivation (2009 to 2013, $n = 417,335$) and Validation cohorts (2009 and 2014, $n = 169,722$) based upon injury index

**Data Availability Statement:** The dataset from this study is held securely in coded form at ICES. While data sharing agreements prohibit ICES from making the dataset publicly available, access may be granted to those who meet pre-specified criteria for confidential access, available at www.ices.on.ca/DAS. The full dataset creation plan and underlying analytic code are available from the DAS (das@ices.on.ca reference TRIM#201609760310000) upon request, understanding that the computer programs may rely upon coding templates or macros that are unique to ICES and are therefore either inaccessible or may require modification.

**Funding:** This study was funded by the Ontario Neurotrauma Foundation (onf.org) MTB, grant number 20160976031000. The funders had no role in the study design, data collection and analysis, decision to publish, or preparation of the manuscript.

**Competing interests:** The authors have declared no competing interests.

**Abbreviations:** AUC, area under the curve; CART, Classification and Regression Tree; DSM-IV, Diagnostic and Statistical Manual of Mental Health Disorders; ED, emergency department; EDHDs, ED visits, hospitalizations, and deaths; ICES, Institute of Clinical and Evaluative Sciences; LOC, loss of consciousness; MOHLTC, Ontario Ministry of Health and Long-Term Care; mTBIs, mild traumatic brain injuries; NACRS, National Ambulatory Care Reporting System; OHIP, Ontario Health Insurance Plan; OMHRS, Ontario Mental Health Reporting System; OR, odds ratio; PPCS, prolonged post-concussion symptoms; PTA, post-traumatic amnesia; TBI, traumatic brain injury; TMJ, temporomandibular joint; TRICORDRR, Toronto Rehabilitation Institute Concussion Outcome Risk Determination and Rehab Recommendations; TRIPOD, Transparent reporting of a multivariable prediction model for individual prognosis or diagnosis; WHO, World Health Organization.

year. Variables selected a priori such as psychiatric disorders, migraines, sleep disorders, demographic factors, and pre-injury healthcare patterns were entered into multivariable logistic regression and CART modeling in the Derivation Cohort to calculate PPCS estimates and forward selection logistic regression model in the Validation Cohort. Variables with the highest probability of PPCS derived in the Derivation Cohort were: Age >61 years ($\hat{p}$ = 0.54), bipolar disorder ($\hat{p}$ = 0.52), high pre-injury primary care visits per year ($\hat{p}$ = 0.46), personality disorders ($\hat{p}$ = 0.45), and anxiety and depression ($\hat{p}$ = 0.33). The area under the curve (AUC) was 0.79 for the derivation model, 0.79 for bootstrap internal validation of the Derivation Cohort, and 0.64 for the Validation model. A limitation of this study was ability to track healthcare usage only to healthcare providers that submit to Ontario Health Insurance Plan (OHIP); thus, some patients seeking treatment for prolonged symptoms may not be captured in this analysis.

## Conclusions

In this study, we observed that premorbid psychiatric conditions, pre-injury health system usage, and older age were associated with increased risk of a prolonged recovery from concussion. This risk score allows clinicians to calculate an individual's risk of requiring treatment more than 6 months post-concussion.

## Author summary

### Why was this study done?

- Between 10% and 20% of adults who are diagnosed with a concussion have symptoms that persist beyond 3 months post-injury.

- The etiology and exact clinical nature of prolonged concussion symptoms remains elusive, and potential predictive factors from the literature remain contentious.

- The goal of the study was to derive a risk score for prolonged post-concussion symptoms (PPCS) among adults with concussion that could enable clinicians treating patients with concussion to determine probable recovery and facilitate appropriate care pathways to improve patient quality of life and recovery.

### What did the researchers find?

- In a cohort of 587,057 adults with a diagnosed concussion, 12.5% met the criteria for PPCS at 6 months following injury.

- Risk of PPCS is highest among those with a pre-injury history of psychiatric disorders and history of anxiety and/or depression.

- Older adults and those with high levels of healthcare usage are at higher risk of developing PPCS.

**What do these findings mean?**

- The risk score may aid physicians treating adults with a concussion by allowing them to quickly assess a patient's risk of prolonged recovery and in turn facilitate tailored treatment plans as appropriate, such as encouraging return to aerobic exercise, education about concussion, timely referrals for specialized psychological care, etc.

- Most adults with a concussion recover by 6 months following injury; however, our results suggest that psychiatric disorders and healthcare utilization are associated with increased risk of PPCS.

## Introduction

Concussions are common, affecting almost 2% of the population at any time [1]. Most symptoms resolve spontaneously in the first few weeks following injury; however, there are highly variable estimates (15% to 40%) [2–4] of the proportion that experience prolonged post-concussion symptoms (PPCS) in the months to years following injury. Numerous factors including, but not limited to, age, sex, headache history, history of mental health problems, loss of consciousness (LOC), and mechanism of injury have been proposed to influence recovery from concussion [5–17]. However, the literature is conflicting and controversial as to which predictive factors affect the development of PPCS in adulthood [18]. The lack of consensus may be due to a number of limitations with previous research including small sample sizes, specific populations studied (e.g., athletes or children), limited duration of follow up, and reliance on self-report measures. In a pediatric concussion population, the Predicting and Preventing Post-Concussion Problems in Pediatrics (5P) study followed individuals presenting to an emergency department (ED) in Canada to determine an absolute risk calculation for persistent concussion-related symptoms at 1 month after injury [19], but no risk score has been developed for adults with concussion.

Health data for each of the more than 13 million residents [20] of Ontario, Canada can be tracked through their unique Ontario Health Insurance Plan (OHIP) number. By linking administrative health databases at the Institute of Clinical and Evaluative Sciences (ICES), cohorts can be created using diagnostic codes, examine healthcare utilization (doctor visits, emergency visits, hospitalizations, wait times, etc.), perform retrospective analyses, and determination of risk [21,22], on the entire province's population. ICES is an independent, non-profit research institute funded by an annual grant from the Ontario Ministry of Health and Long-Term Care (MOHLTC). As a prescribed entity under Ontario's privacy legislation, ICES is authorized to collect and use healthcare data for the purposes of health system analysis, evaluation, and decision support. Secure access to these data is governed by policies and procedures that are approved by the Information and Privacy Commissioner of Ontario.

The objective of this study was to derive a risk score for PPCS by determining which demographic factors, premorbid health conditions, and healthcare utilization patterns are associated with PPCS using a large cohort of adults with concussion identified using administrative health databases housed at ICES. We hypothesized that mental health issues, headache, and neurological issues would be associated with increased incidence of PPCS and that it would be feasible to derive a PPCS risk score using available information.

## Methods

### Ethics statement

This study was approved by the Research Ethics Board at University Health Network (ID# 15–9532). The data were anonymized, reported in aggregate, and cells with less than 6 were suppressed as per ICES Privacy Policy. Consent was waived as ICES is a prescribed entity under section 45 of Ontario's Personal Health Information Protection Act. Section 45 authorizes ICES to collect personal health information, without consent, for the purpose of analysis or compiling statistical information with respect to the management of, evaluation or monitoring of, the allocation of resources to or planning for all or part of the health system. This study is reported as per the Transparent reporting of a multivariable prediction model for individual prognosis or diagnosis (TRIPOD) statement (S1 Checklist).

### Setting

Ontario had a population of 13,448,949 in the 2016 Census [20] and is the most populous province in Canada. It has the second largest geographical area. Approximately 15% of the population resides in areas classified as rural (defined by Statistics Canada as not residing in a metropolitan area or its surrounding area). The average age of residents of Ontario in the 2016 Census was 41.0 (no standard deviation was provided), and the median was 41.3 years. Females comprised 50.6% of the population. Visible minorities, as defined by Statistics Canada as "persons, other than Aboriginal peoples, who are non-white in colour," accounted for 29.3% of the population. English is the most commonly spoken language (97.3%).

### Data sources

Records were identified in ICES through the (1) National Ambulatory Care Reporting System (NACRS), for people diagnosed in an ED using ICD-10 diagnostic codes, or (2) the OHIP Physician Billing Database, for records containing diagnoses by a primary care physician using ICD-9 diagnostic codes. OHIP is a taxpayer-funded health insurance plan for all eligible residents of Ontario. Physicians, laboratories, and diagnostic facilities submit the patient's OHIP identifier, type of visit, diagnosis, tests performed, etc. to OHIP to receive payment. All follow-up visits were linked using the individuals' unique OHIP identifier and abstracted from the physician billing database. The premorbid health retrospective review was conducted using the relevant ICD-9 diagnostic codes (S1 Table) 5 years prior in the OHIP Physician Billing Database for each linked OHIP identifier and, additionally, included the Ontario Mental Health Reporting System (OMHRS) database for psychiatric hospital admissions. These datasets were linked using unique encoded identifiers and analyzed at ICES.

### Study cohorts

The cohort consisted of adults ≥18 years of age with a valid OHIP number and a physician-diagnosed concussion between 2008 and 2014. Cases with injury index dates in 2015 and 2016 were excluded due to delay in OHIP reporting to ICES that did not permit 2 years healthcare tracking after index date. ICD-9 code 850 was used to identify concussions diagnosed by primary care physicians, and ICD-10 S06 was used to identify those diagnosed in the ED. Duplicate records, such as those with all variables identical or if multiple records with the same diagnostic billing code per person per physician per day (only 1 record will be counted), were removed. Records were excluded if there were incomplete demographic data, the individual was not a resident of Ontario, age was greater than 105 years, or if a date of death was listed before the index event date. There was a 365-day washout period for previous traumatic brain

injuries (TBIs) to ensure that the cohort would not have duplication of identified index cases. To ensure cohort data contained only mild traumatic brain injuries (mTBIs), records of anyone with admission to hospital or death within 30 days of the index event were excluded (Fig 1).

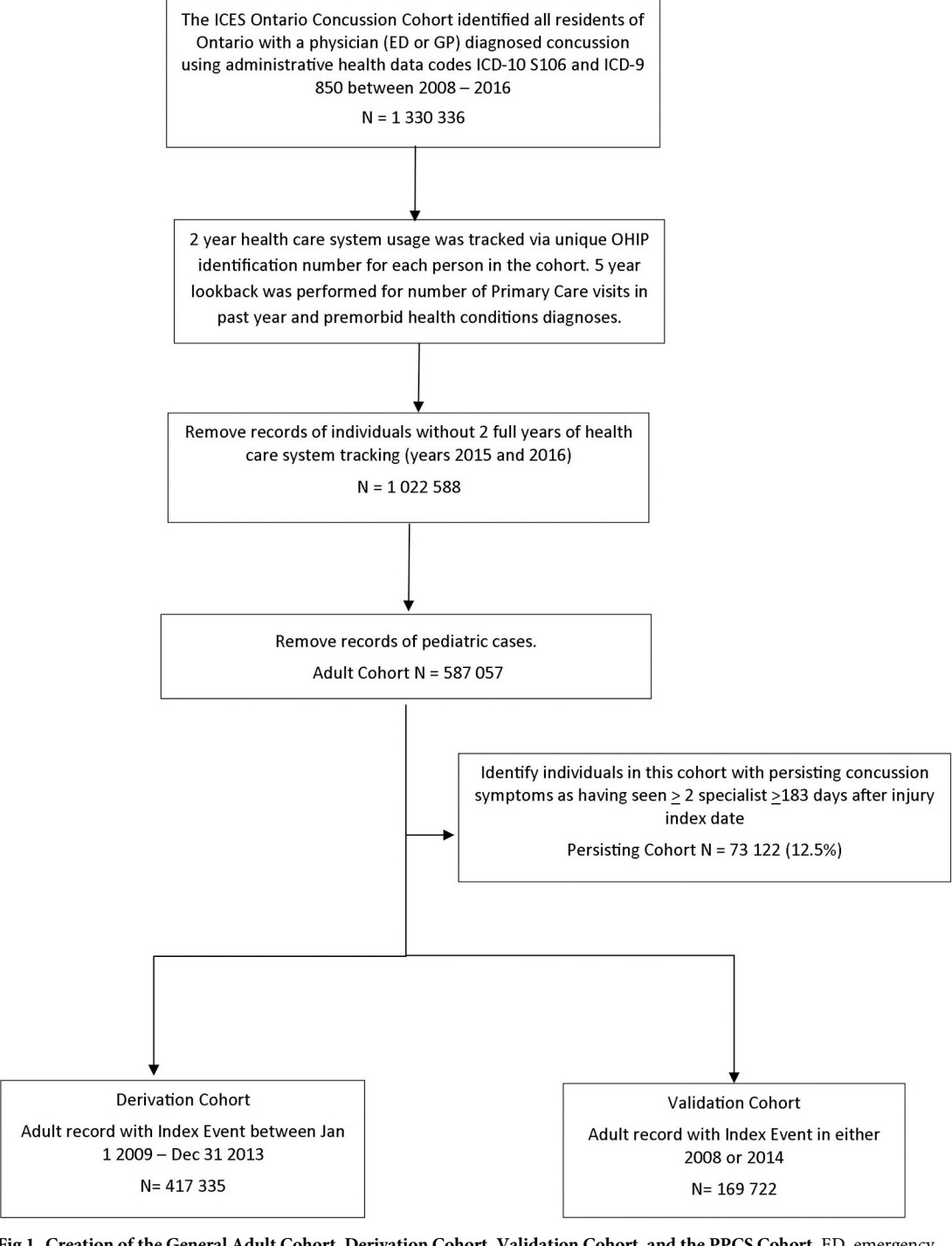

**Fig 1. Creation of the General Adult Cohort, Derivation Cohort, Validation Cohort, and the PPCS Cohort.** ED, emergency department; GP, general practice; ICES, Institute of Clinical and Evaluative Sciences; OHIP, Ontario Health Insurance Plan; PPCS, prolonged post-concussion symptoms.

This concussion cohort was tracked for 2 years after their index date of injury (the date recorded by the physician at the initial point of access to the healthcare system associated with concussion diagnosis code) using their unique OHIP identifier for concussion-related follow-up care in the OHIP Physician Billing Database. Specialists identified for providing relevant care were as follows: neurosurgery, neurology, physical medicine, otolaryngology, ophthalmology, and psychiatry. Next, a 5-year retrospective review of frequency of premorbid healthcare system utilization and relevant premorbid diagnoses was performed. Diagnoses of interest were selected based on a literature review of the following predictive factors for prolonged concussion recovery: depression and anxiety, personality disorders, bipolar disorder, migraine, headache without migraine, sleep disorders, pain disorders, neurological disorders (multiple sclerosis, dementia, stroke, etc.), vestibular disorders (i.e., ataxia, gait disorders, dizziness not caused by a primary disease like hypotension or a neurological disorder), temporomandibular joint (TMJ) disorders, schizophrenia, alcohol misuse disorder and illicit drug use disorders, and prior TBIs, using the OHIP Physician Billing Database and OMHRS for schizophrenia, personality disorders, and bipolar disorder (S1 Fig). The diagnostic billing codes used are available in S1 Table.

PPCS does not have a validated or unanimously agreed upon definition for diagnosis or even accepted terminology. The World Health Organization (WHO) [23] criteria for post-concussion syndrome is 3 out of 8 possible symptoms (i.e., headache, dizziness, fatigue, irritability, insomnia, concentration or memory problems, intolerance of stress, emotion, or alcohol) within 30 days of a head injury sufficient to result in LOC. Criteria from the Diagnostic and Statistical Manual of Mental Health Disorders (DSM-IV) [24] included neuropsychological impairments as well as symptoms prolonged more than 3 months post-injury that were not present pre-injury or worsened post-injury. For the purpose of this analysis, PPCS was defined as individuals with concussion requiring assessment by 2 or more specialists appropriate for concussion-related symptoms (neurology, otolaryngology, psychiatry, neurosurgery, etc.) more than 182 days (6 months) after their injury index date. Six months was selected to allow patients with prolonged symptoms to access a primary care physician in order to obtain a referral for a specialist physician, which may have long wait times for a first visit.

The "Derivation Cohort" was composed of all adults diagnosed with a concussion between the years 2009 and 2013, and all analyses to create the predictive model were performed on this cohort. The "Validation Cohort" included all adults diagnosed with a concussion in either 2008 or 2014 and was used to validate the predictive model. The year 2008 was selected as it was assumed to have lower rates of referrals due to lower physician awareness on the topic of concussion. The year 2014 was selected due to increased awareness in the medical community due to increased research as well as high-profile injuries to athletes and military personnel that we believe would have resulted in higher rates of referrals to specialist physicians. Adults with an index date in either 2015 or 2016 did not have full 2 years of specialist healthcare tracking and were not included in the risk score analysis.

## Analyses

An analysis plan was developed as part of the Dataset Creation Plan prior to initiating the study and included the univariate logistic regressions used to determine the variables entered into a stepwise multivariate regression model and then a multivariate logistic regression model in the Derivation Cohort, forward selection multivariate logistic regression model in the Validation Cohort, and bootstrapping of the Derivation Cohort. The Classification and Regression Tree (CART) and derivation of a risk score were prompted by the data analysis after the multivariate logistic regression model. Secondary descriptive analyses comparing the Derivation and Validation Cohorts were requested by reviewers. (S1 Text) All statistics were performed

on SAS 9.4 (SAS Institute, Cary, North Carolina, USA) for Windows. Nineteen variables were selected a priori for analysis. Potential covariates were sex, age group, premorbid health conditions, frequency of primary care use in the year previous to the injury index date, and rurality (not residing in a metropolitan area). Univariate logistic regressions were performed between each premorbid condition, sex, and age group and the number of specialist visits in the persistent post-concussion symptoms cohort. Linear regressions were performed between the number of primary care visits the year before the injury and specialist follow-up visits. Chi-squared analysis of rurality classification of the record and inclusion in the Prolonged Cohort was performed. Variables with statistically significant ($p < 0.05$) odds ratio (OR) or significant linear regressions were entered into a stepwise regression model. Variables from the univariate analysis were retained in the model if they had a $p$-value less than 0.01 and underwent stepwise regression. Only variables that had a significant $p$-value less than 0.001 in the stepwise regression were retained for the multivariate logistic regression model. The variables that remained significant were entered into a CART analysis using the derivation concussion cohort with a diagnosis made between 2009 and 2013. Forward validation of the model was performed on the validation cohort with an index date in 2008 or 2014 with the same variables retained from the multiple covariate logistic regression in the derivation cohort. Risk score calculations were made for the significant nodes of the CART analysis using the methods described by Sullivan and colleagues [25] for deriving risk scores using the results from a multivariate logistic regression model on a binary outcome. The final model using variables retained from the multivariate logistic regression was validated internally in the Derivation Cohort (2009 to 2013) using bootstrap resampling of 1,000 randomly selected samples.

## Results

The records of 1,567,508 individuals were initially identified. A total of 197,489 duplicate records were excluded, and 39,683 records were excluded for insufficient demographic data or potentially more severe injuries. The final Ontario Concussion Cohort totaled 1,330,336 records. This cohort was restricted to those with 2 years post-injury follow-up of healthcare surveillance (index years 2008 to 2014) and were 18 years of age or older at the index date resulting in 587,057 cases. Of this, 73,122 people (12.5%) met our criteria for PPCS (The Prolonged Cohort). The demographics are presented in Table 1. The "Derivation Cohort" was composed of 417,335 records. The "Validation Cohort" contained 169,722 records. Due to the high rates of psychiatric healthcare use prior to their injury, schizophrenia, and illicit drug use were not further analyzed in this model. Prevalence of the premorbid conditions is presented in Table 2.

Overall, there were 248,765 (42.4%) females and 338,292 (57.6%) males included in the analysis. There were 35,344 (48.3%) females and 37,778 (51.7%) males in the Prolonged Cohort. Approximately 80% had a concussion diagnosed by an ED physician.

The results from the univariate logistic regressions are presented in S2 Table. There was a significant relationship ($p < 0.0001$, unadjusted $r^2 = 0.265$) between the number of primary care visits the year prior to the index concussion and requiring specialist visits for PPCS. The mean number of primary care visits for the Prolonged Cohort in the year prior to concussion was 14.0 visits with a standard deviation of 18.7; the median was 9 visits (range 0 to 365). Quartiles for primary physician visits in the Derivation Cohort were 0 to 4 visits (low usage), 5 to 8 visits (average usage), 9 to 14 visits (above average usage), and >15 visits the previous year (high usage); the categories were used for all later analyses instead of continuous value. There was no relationship between living in either a rural or urban/suburban area of Ontario and meeting prolonged cohort criteria ($\chi^2 = 0.16$, $p = 0.69$); however, those with a rural

**Table 1. Demographics of the overall Adult Concussion Cohort (2008–2014) and subcohorts of Prolonged Symptoms (2008–2014), the Derivation Cohort (2009–2013), and the Validation Cohort (2008 and 2014).**

| | | 2008–2014 Adult Concussion Cohort | Adult Prolonged Symptoms Cohort | Derivation Cohort (2009–2012) | Validation Cohort (2008 and 2014) |
|---|---|---|---|---|---|
| Total | | 587,057 | 73,122 | 417,335 | 169,722 |
| Sex (%) | Female | 248,765 (42.4%) | 35,344 (48.3%) | 175,962 (42.2%) | 72,803 (42.9%) |
| | Male | 338,292 (57.6%) | 37,778 (51.7%) | 241,373 (57.8%) | 96,919 (57.1%) |
| Age group (%) | 18–30 | 181,811 (31.0%) | 13,544 (7.4%) | 130,202 (31.2%) | 51,609 (30.4%) |
| | 31–40 | 84,587 (14.4%) | 8,530 (10.0%) | 59,903 (14.4%) | 24,684 (14.5%) |
| | 41–50 | 88,463 (15.1%) | 11,157 (12.6%) | 62,937 (15.1%) | 25,526 (15.0%) |
| | 51–60 | 75,741 (12.9%) | 11,483 (15.2%) | 53,396 (12.8%) | 22,345 (13.2%) |
| | 61–80 | 100,703 (17.2%) | 19,979 (19.8%) | 71,399 (17.1%) | 29,304 (17.3%) |
| | >81 | 55,752 (9.5%) | 8,429 (15.1%) | 39,498 (9.5%) | 16,254 (9.6%) |
| Diagnosed (%) | ED | 483,051 (82.3%) | 59,172 (80.9%) | 343,084 (82.4%) | 139,084 (82.0%) |
| | GP physician | 104,006 (17.7%) | 13,950 (19.1%) | 73,368 (17.6) | 30,638 (18.0%) |
| Mean GP visits 1 year before index date (SD) | | 14.0 (SD 18.7) | 19.1 (SD 20.6) | 13.9 (SD 18.6) | 14.2 (SD 18.7) |
| Median GP visits 1 year before index date (range) | | 9 (0–365) | 13 (0–365) | 9 (0–365) | 9 (0–365) |

ED, emergency department; GP, general practice.

classification had a longer wait to access care (rural mean 213.5 (SD 196.4) days after index, nonrural mean 202.3 (SD 197.0) days after index, $p < 0.0001$). The location of diagnosis (ED or primary care) was not significantly associated with the number or type of specialist visits for prolonged concussion-related symptoms (OR 1.00, CI 0.996 to 1.006, $p = 0.83$). All covariates

**Table 2. Prevalence of premorbid conditions in the Adult Concussion Cohort and the Adult Prolonged Symptoms, Derivation Cohort (2009–2013), and the Validation Cohort (2008 and 2014) subcohorts.**

| Premorbid factor | | 2008–2014 Adult Concussion Cohort (% prevalence in cohort) | Derivation Cohort (2009–2013) | Validation Cohort (2008 and 2014) | Adult Prolonged Symptoms Cohort (% prevalence in cohort) | Percent with prolonged symptoms |
|---|---|---|---|---|---|---|
| Mental health | Anxiety and depression | 244,045 (41.6%) | 172,521 (41.3%) | 71,524 (42.1%) | 34,856 (47.7%) | 14.3% |
| | Personality disorders | 12,204 (2.1%) | 8,557 (2.1%) | 3,647 (2.2%) | 1,856 (2.5%) | 15.2% |
| | Bipolar disorder | 1,627 (0.27%) | 1,098 (0.26%) | 529 (0.31%) | 233 (0.32%) | 14.3% |
| | Schizophrenia | 12,801 (2.2%) | 9,091 (2.2%) | 3,170 (2.2%) | 1,772 (2.4%) | 13.8% |
| | Illicit drug use | 34,443 (5.9%) | 24,386 (5.8%) | 10,057 (5.9%) | 4,980 (6.8%) | 14.5% |
| | Other | 74,036 (12.6%) | 52,405 (12.6%) | 21,631 (12.7%) | 11,321 (15.5%) | 15.3% |
| Neurological disorders | | 22,082 (3.8%) | 15,545 (3.7%) | 6,537 (3.9%) | 3,312 (4.5%) | 15% |
| Sleep disorders | | 179,310 (30.5%) | 126,844 (30.4%) | 52,466 (30.9%) | 26,788 (36.6%) | 15% |
| Pain disorders | | 146,168 (24.9%) | 103,323 (24.8%) | 42,845 (25.2%) | 21,759 (29.8%) | 14.9% |
| Migraine | | 36,722 (6.3%) | 25,405 (6.2%) | 10,767 (6.3%) | 5,401 (7.4%) | 14.7% |
| Headache without migraine | | 40,083 (6.8%) | 28,372 (6.8%) | 11,711 (6.9%) | 5,810 (7.9%) | 14.5% |
| Vestibular/balance disorders | | 348,185 (59.3%) | 246,986 (59.2%) | 101,199 (59.6%) | 49,012 (67.0%) | 14.1% |
| TMJD | | 4,371 (0.7%) | 3,059 (0.7%) | 1,312 (0.8%) | 637 (0.9%) | 14.6% |
| Prior concussion/TBI | | 25,520 (4.3%) | 17,349 (4.2%) | 8,171 (4.8%) | 3,273 (4.5%) | 12.8% |

TBI, traumatic brain injury; TMJD, temporomandibular joint dysfunction.

**Table 3. Multivariate logistic regression in the Derivation Cohort (2009–2013).**

| | | OR | 95% confidence interval | p-value |
|---|---|---|---|---|
| Mental health | Bipolar disorder | 4.28 | 3.86–4.76 | <0.0001 |
| | Personality disorder | 2.40 | 2.31–2.50 | <0.0001 |
| | Anxiety/depression | 1.32 | 1.31–1.34 | <0.0001 |
| | Other | 1.34 | 1.31–1.37 | <0.0001 |
| Number of primary care visits year prior | >15 | 5.51 | 5.41–5.62 | <0.0001 |
| | 8–14 | 3.04 | 2.99–3.10 | <0.0001 |
| | 5–7 | 1.92 | 1.89–1.97 | <0.0001 |
| | 0–4 (reference group) | | | |
| Sex (female) | | 0.95 | 0.94–0.97 | <0.0001 |
| Age group | 18–30 | 0.30 | 0.29–0.36 | <0.0001 |
| | 31–40 | 0.35 | 0.35–0.36 | <0.0001 |
| | 41–50 | 0.49 | 0.48–0.51 | <0.0001 |
| | 51–60 | 0.69 | 0.68–0.71 | 0.00002 |
| | 61–80 | 1.63 | 1.62–1.72 | <0.0001 |
| Neurological disorders | | 1.76 | 1.70–1.81 | <0.0001 |
| Pain disorders | | 1.26 | 1.21–1.30 | <0.0001 |
| Migraine | | 1.26 | 1.23–1.29 | <0.0001 |
| Sleep disorders | | 0.95 | 0.92–0.99 | 0.0001 |
| Vestibular disorders | | 1.47 | 1.45–1.49 | <0.0001 |
| TMJD | | 1.14 | 1.05–1.19 | 0.001 (NS) |

NS, not significant; OR, odds ratio; TMJD, temporomandibular joint dysfunction.

except for rurality and ED/ primary care diagnosis were entered into the stepwise model (S2 Table). Prior TBI ($p = 0.21$) and headache without migraine ($p = 0.86$) were not retained in the stepwise model. TMJ disorders were not retained in the multivariate logistic regression model ($p = 0.001$). Sex was also excluded due to small effect size (effect 0.001) despite reaching the significance threshold ($p < 0.0001$) (Table 3 and Fig 2). The intercept of the multivariate logistic regression model was −0.92. The area under the curve (AUC) for this model was 0.79 (95% CI 0.78 to 0.80) (Fig 3), and when the model was internally validated by bootstrap resampling, the AUC was 0.79 (95% CI 0.77 to 0.81).

Variables entered into the CART analysis with outcome of number of specialist visits included sex, age group, healthcare use frequency, personality disorders, bipolar disorder, depression and anxiety, migraine, pain disorders, neurological disorders, sleep disorders, vestibular disorders, and other mental health disorders. Personality disorders were the primary node in this analysis; however, given the low prevalence of personality disorders in the general population and difficulty of reliable diagnosis, personality disorders and bipolar disorder were removed and a second CART was performed (Fig 4). The significant nodes (99.5%) of the CART were bipolar disorder, personality disorder, high number of primary care appointments the year prior, premorbid diagnosis of depression and anxiety, and age group. Those with premorbid pain or neurological disorders also had the greatest average number of specialist visits (9) greater than 6 months after their injury, although these nodes were not significant for prediction. Forward model validation in the Validation Cohort included retained variables primary care usage, age group, vestibular disorders, personality disorders, anxiety and depression, neurological disorders, other mental health disorders, migraine, and bipolar disorders and accounted for 99.6% of the model (S2 Fig) with an AUC of 0.64 (95% CI 0.63 to 0.65) (Fig 5).

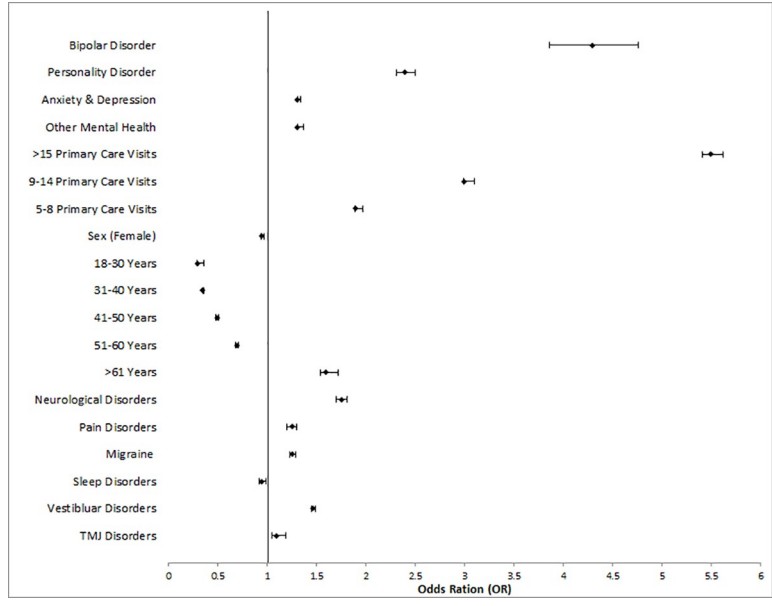

**Fig 2. Multivariate logistic regression results in Derivation Cohort with OR and 95% confidence intervals.** The reference group for primary care visits comparisons was 0–4 primary care visits. Value are also presented in Table 2. OR, odds ratio; TMJ, temporomandibular joint disorders.

Risk modifiers were bipolar disorder (52% risk), personality disorder (45%), more frequent than monthly primary care use (46%), 9 to 12 primary care visits in the year prior (34%), 5 to 8 primary care visits the year prior (26%), 0 to 4 primary care visits the year prior (20%), preexisting anxiety or depression (33%), 61-plus years (55%), 51 to 60 years (23%), 41 to 50 years (20%), 31 to 40 years (17%), and 18 to 30 years (15%). For ease of use by clinicians, the raw estimates for each variable were converted to a point value, and a risk point system to estimate risk score for adults requiring 2 or more specialist visits more than 6 months after their concussion in the Derivation Cohort was determined using the method detailed by Sullivan and colleagues [25] (Table 4). The Toronto Rehabilitation Institute Concussion Outcome Risk Determination and Rehab Recommendations (TRICORDRR) risk calculator tool is available in S1 Tool TRICORDRR and online (https://kite-uhn.com/tricordrr). A hypothetical patient aged 50 to 60 years (0 point) with 6 primary care visits in the year prior (0 point), history of depression (1 point) but no other psychiatric history (0 point) would have a total point score of 1 and corresponding risk of 34%. When risk for PPCS is calculated using the equation containing the raw values for each variable instead of the point system, the risk is 31.6%. A 20-year-old (−3) with no primary care visits the year before (−1) and no mental health history would have a point of −4, corresponding risk of 9%, and a raw risk of 11.8%. A patient in their 60s (3) with more than monthly primary care visits (4) and diagnosis of anxiety (1) would have a point score of 8 corresponding to a risk of 78% and a raw calculated risk of 76.2%.

The proportion of cases that had PPCS for each point score strata was identified using the Validation Cohort (Fig 6). The proportion that actually met criteria was determined by dividing the PPCS cases by total number of patients who had that specific point score. Each risk stratum is comprised of multiple subgroups (i.e., individuals can be assigned to that strata based on different combinations of points representing their individual premorbid and demographic factors). More than 51% of the Validation Cohort scored a combined risk score of 0 or lower on the TRICORDRR.

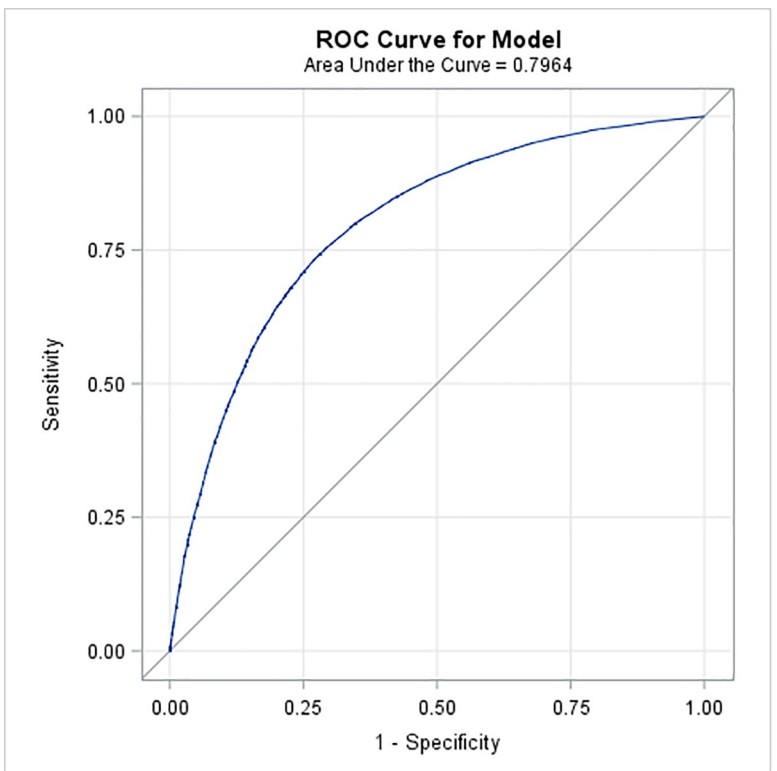

**Fig 3. AUC of Derivation Cohort 0.79 (95% CI 0.78–0.80).** It has 75% sensitivity and 70% specificity. AUC, area under the curve; ROC, receiver operating characteristic.

## Discussion

This study observed that 12.5% of adults diagnosed with a concussion would require specialized medical care related to their injury 6 months or longer post-injury. Older age (over 61 years), premorbid psychiatric and mental health history, and high usage of primary care in the year before injury were associated with greater risk of PPCS. To the authors' knowledge, this is the largest study to date, with almost 600,000 adults, examining predictive factors for PPCS requiring specialized healthcare. The Ontario Concussion Cohort is also the second largest concussion cohort globally; only the CDC's TBI-related ED visits, hospitalizations, and deaths (EDHDs) cohort is larger at 2.87 million people [26], although it also included moderate to severe TBIs. The size of this study enabled both the estimate of the frequency of PPCS and development of a risk stratification model.

### Prolonged symptoms cohort

Approximately 50.3% of all cases included in this analysis visited a specialist in the 2 years after the index injury date for either a concussion or concussion-related symptom. The Prolonged Symptom Cohort consisted of 73,122 cases, 12.5% of the overall cohort, which met the criteria of >2 specialist visits >6 months after the index injury (12.4% of the Derivation Cohort met criteria, and 12.7% of the Validation Cohort met criteria). This is consistent with estimates suggesting that 10% to 20% of adults with a concussion continue to experience symptoms more than 3 months post-injury [2,3]. This is based on the presence of a diagnosis of concussion or concussion-related symptoms (e.g., headache, migraine, poor mental health) by physicians in Ontario (although there is no billing code in Ontario for "prolonged post-concussion

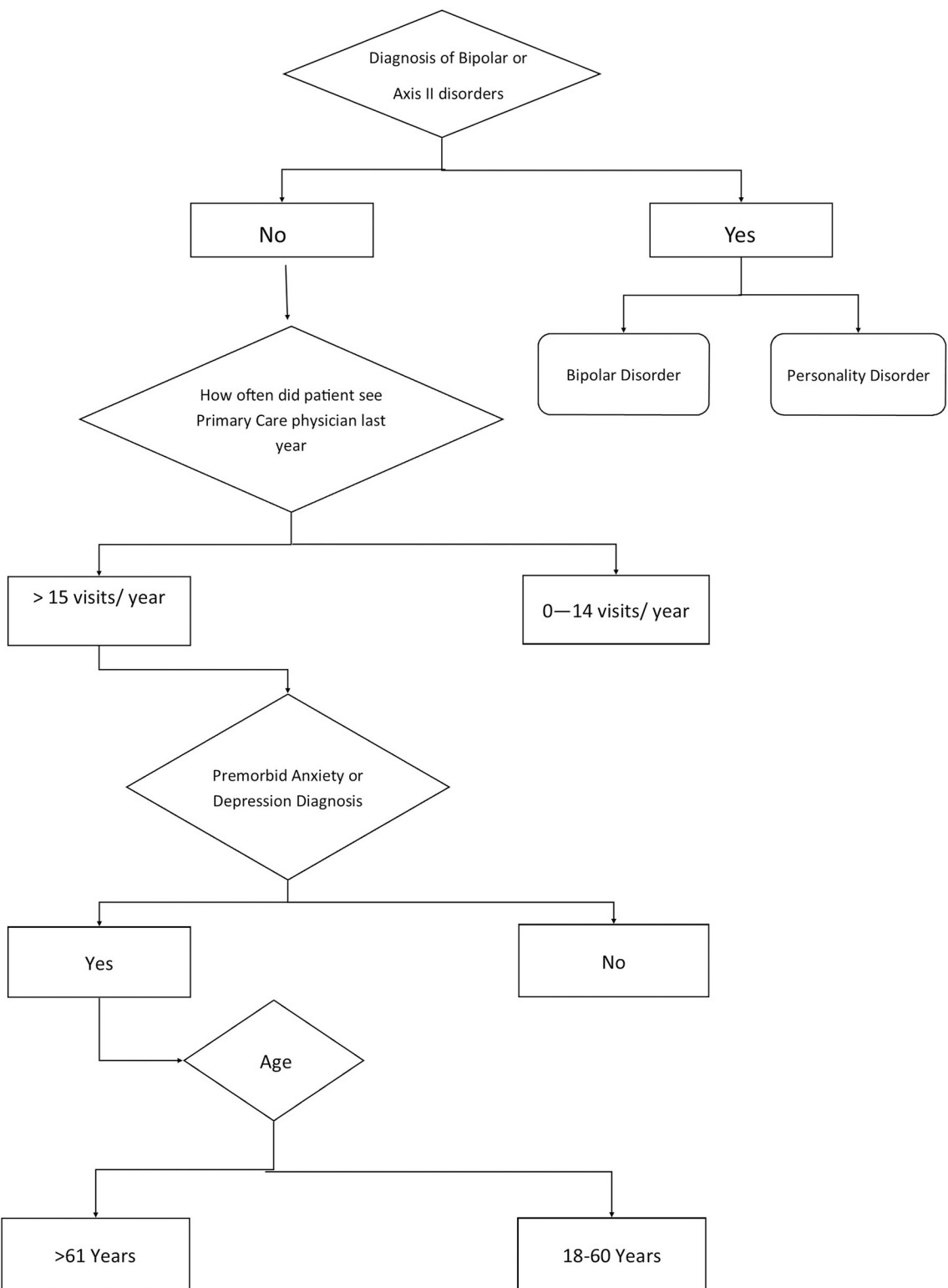

**Fig 4. Decision rule tree from Derivation Cohort CART.** Terminal nodes with the highest risk of PPCS are for individuals with personality disorders or bipolar disorder or adults over 61 years with high levels of primary care usage in the year before injury with a history of anxiety and/or depression. CART, Classification and Regression Tree; PPCS, prolonged post-concussion symptoms.

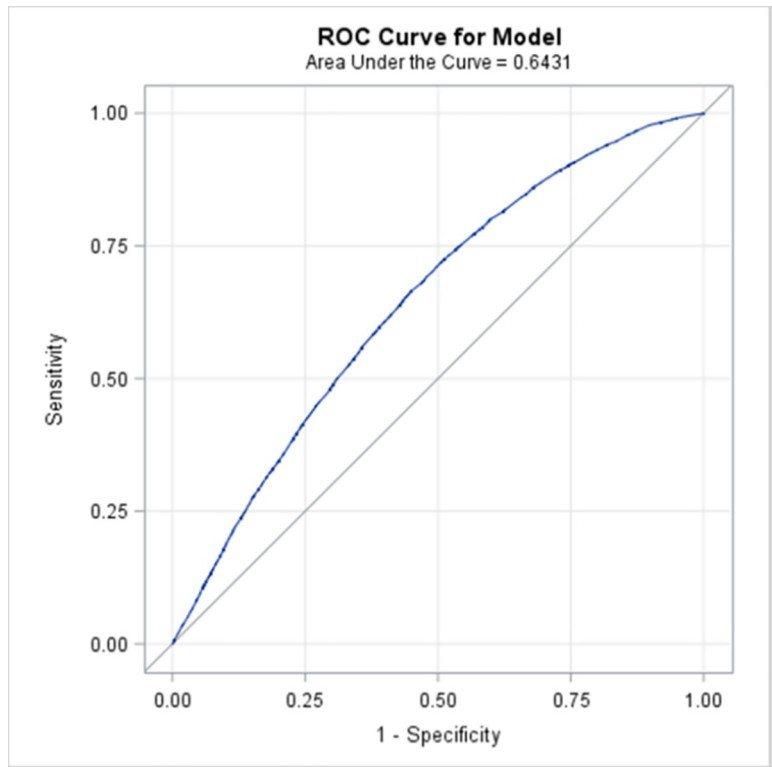

**Fig 5. AUC of Validation Cohort 0.64 (95% CI 0.63–0.65).** The validated model has a sensitivity of 75% and a specificity of 47%. AUC, area under the curve; ROC, receiver operating characteristic.

symptoms"). We identified people who were symptomatic enough to require a specialist more than 6 months post-injury from the prolonged symptom cohort. When criteria for PPCS was relaxed to identify those with specialist visits 3 months after index date, 175,135 cases (29.8%) were identified, which is also consistent with estimates of PPCS. However, only 72.2% of cases had follow-up by a primary care physician within 3 months of injury, and some of the visits may have been for pre-injury referrals depending upon the wait time(s) to see that particular specialist. The 12.5% PPCS is a more conservative estimate that most likely reflects those cases with true prolonged symptoms.

## Risks associated with mental health conditions

The most predictive factors in developing PPCS were a prior diagnosis of mental health problems, especially depression, anxiety, bipolar and personality disorders, high frequency of pre-injury primary healthcare use, and age. Consistent with the literature, mental health and psychiatric premorbid diagnoses were associated with greater healthcare use for PPCS [14–17]. Personality disorders and bipolar disorders have the greatest predictive risk for developing PPCS. People with these diagnoses were identified in OMHRS such that the diagnosis was made by either a psychiatrist or a psychologist, and these individuals would have received some form of inpatient treatment for their mental health. Personality disorders in particular have been found to be associated with increased healthcare utilization independent of physical health [27,28]. Bipolar disorder was the most significant predictive factor for developing PPCS, although it is extremely rare, difficult to diagnose, has high similarity with other psychiatric disorders, and most clinicians will probably encounter few in their practice. Furthermore,

**Table 4. Factor specific risk scores of the Derivation Cohort and estimated risk of the combined point for an individual.**

| | | Point value |
|---|---|---|
| Bipolar disorder | | 4 |
| Personality disorder | | 3 |
| Anxiety and depression | | 1 |
| Primary care visits year prior to injury | >15 | 3 |
| | 9–14 | 1 |
| | 5–8 | 0 |
| | <4 | −1 |
| Age group | 18–30 | −3 |
| | 31–40 | −2 |
| | 41–50 | −1 |
| | 51–60 | 0 |
| | >61 | 4 |

| Combined point score | Estimated risk |
|---|---|
| −4 | 9% |
| −3 | 14% |
| −2 | 19% |
| −1 | 23% |
| 0 | 28% |
| 1 | 34% |
| 2 | 44% |
| 3 | 47% |
| 4 | 54% |
| 5 | 61% |
| 6 | 67% |
| 7 | 73% |
| 8 | 78% |
| 9 | 82% |
| 10 | 86% |
| 11 | 89% |
| ≥12 | 91% |

This study was approved by the Research Ethics Board at the University Health Network.

OHIP billing code for bipolar disorder does not differentiate between subcategories of the disorder. Those with these diagnoses typically have frequent primary care visits and comorbid conditions like illicit drug use disorders and other mental health conditions [29].

Anxiety and depression, both premorbid and after injury, have been consistently identified as predictive factors of PPCS in adult and pediatric populations. Anxiety in particular has been shown to influence severity of symptoms of acute concussion and has mediating effects on post-injury catastrophization [30], somatoform disorders, and other post-concussion symptoms like sleep disorder, pain, and headache.

## Prior healthcare utilization

High frequency of primary care utilization in the year prior to injury as a predictor of prolonged symptoms is a novel finding for concussion but has been found in other conditions such as spinal pain [31]. There are a number of potential reasons including premorbid health

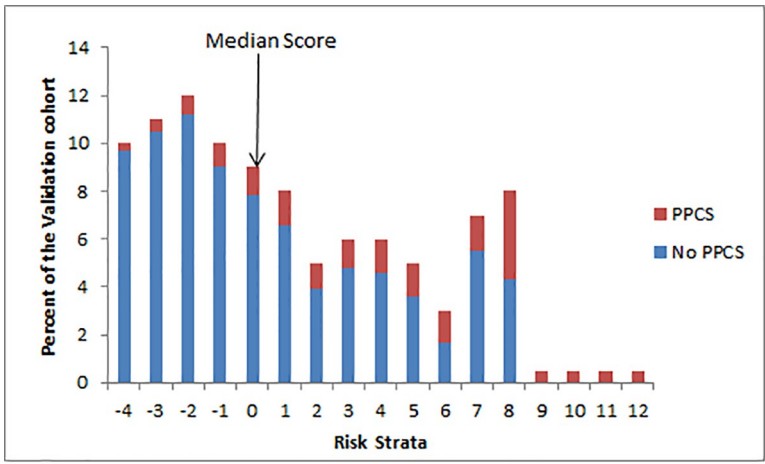

**Fig 6. Identification of cases that met PPCS criteria in the Validation Cohort. Median risk strata was 0. PPCS, prolonged post-concussion symptoms.**

concerns or healthcare-seeking behaviors, which may be mediated by factors such as sex, education, health literacy, premorbid coping strategies, a tendency to catastrophize, and somatoform disorders [32]. Interestingly, the mean number of primary care visits associated with premorbid health conditions like mental health and migraine did not differ statistically significantly from the overall cohort, so this finding may indeed be a result of healthcare-seeking behaviors, somatization, and innate coping strategies.

## Age-related risks

Recovery from concussion has been shown to vary based upon the individual's age [7,8]; however, older adults are rarely studied in concussion research, making the increased risk of PPCS in seniors a potentially novel finding. This study suggests that younger adults are more likely than their older counterparts (age >61) to recover post-concussion. This may relate to more prevalent premorbid health conditions in the older group (as shown with a significantly higher ($p < 0.0001$) number of primary care visits) than in younger age groups. It might also relate to reduced neurocognitive reserve capacity in the older group. Neurocognitive reserve capacity is a proposed protective mechanism that may enable some individual's brain to adapt to or recover from pathology associated with neurological diseases and/or psychiatric disorders and is believed to decrease with the aging process [33]. Indeed, there has been some preliminary evidence to suggest that lower cognitive reserve is associated with worse cognitive outcome post-mTBI [34] as well as risk for persistent post-concussive symptoms [35].

## Model fit

The model fit in all models was moderate though lower in the Validation Cohort. This may be due to the years selected a priori for the Validation Cohort. The years included for the Validation Cohort were selected for clinical reasons. Concussion awareness was assumed to be lower in 2008 than in 2014 [1] among both healthcare providers and the general public because there were more peer-reviewed papers published in 2014 than 2008 as indexed by PubMed; the release of treatment guidelines (e.g., Ontario Neurotrauma Foundation Guidelines for Concussion/Mild Traumatic Brain Injury and Persistent Symptoms in 2013 [36]; updates on the Consensus Statement on Concussion in Sport that were published in 2008 [37] and 2012 [38])

was assumed to have increased adherence to best practices in 2014, and, also, there were high-profile concussions among professional athletes. The 2 years (2008 and 2014) were adopted to balance the low and high awareness of concussion at either end of the cohort timeline. The proportion of cases that were referred for additional healthcare for PPCS was higher in 2014 than 2008 (13.1% in 2014 cf. 12.3% in 2008). Furthermore, there were fewer diagnosed concussions in 2014 than 2008, and the decrease in cases was not offset by the increase in people meeting the PPCS criteria, particularly in age groups under 51 years as the Validation Cohort had slightly greater proportions of adults in older age groups than the Derivation Cohort, but these differences were not statistically significant (range $p$ = 0.45 to 0.62). This discrepancy may have affected accuracy of the risk estimates, particularly in lower-risk younger age groups without a psychiatric history. There was less concordance between the observed prevalence for each point estimate between the Derivation and Validation Cohorts particularly of risk score of 1 to 5 points, and this may due to the multiple possible combinations of risk factors (i.e., age group, frequency of primary care usage, and mental health status that may generate that point score).

## Other potential risk modifiers

The branches on the CART analysis for premorbid neurological disorders were below the threshold for significance, and, therefore, neurological disorders were not included in the formal risk score; however, these conditions are important clinically as they have the highest number of visits with specialist physicians compared with other premorbid conditions. This may be due to decreased neurocognitive reserve associated with the neuropathology of these disorders. The additional neurological burden resulting from the concussions may further exasperate premorbid symptoms associated with their neurological disorders.

The literature is inconsistent for the role of confounding variables like migraine history, sex, headache, and sleep disorders in developing PPCS. In this model, although some retained significance through the multivariate regression step, they were not retained in the CART analysis and were not included in the final risk score calculation. There are other prospective cohort studies that are also analyzing possible factors for risk of developing PPCS such as TRACK-TBI or the NCAA cohort. A Nelson and colleagues study [39] included all people with mild TBI at level 1 trauma centers including some who were CT scan positive. They acknowledge that this may not reflect the broader mTBI population; however, they clearly identify that a significant proportion 53% were having difficulties at 12 months. A study by Stein and colleagues [40] also confirmed our findings that those with history of preexisting mental health issues were at higher risk of developing complications from their mild TBI. This study also included 66% who were severe enough for admission to hospital and 24.5% admitted to ICU. This study involves much more severe TBI than our cohort. Risk factors for probable depression at 6 months after mTBI included less education (adjusted OR, 0.89; 95% CI, 0.82 to 0.97 per year), being Black, self-reported psychiatric history, and injury resulting from assault or other violence. Unfortunately, no race-based or individual educational data were available. The Stein group did not calculate a risk score, although they clearly identified similar risk factors to our study; however, their more severely injured population precludes direct comparison to our study group.

## Future directions

The TRICORDRR tool is currently being prospectively validated by staff physicians at the Hull-Ellis Concussion and Research Clinic at Toronto Rehabilitation Institute–University Health Network.

## Limitations

We acknowledge some limitations. For example, we were unable to verify the diagnoses of the included cases in this study in the medical charts and are reliant upon doctors recording the correct diagnostic billing code. As there is also no currently accepted definitive diagnostic criteria or billing code for "prolonged post-concussion symptoms," we had to create our own criteria to identify those most likely to be seeking medical treatment for PPCS. Furthermore, sports medicine physicians are classified as primary care physicians by their OHIP specialty code, and, therefore, the patients they treated are not included in this analysis of specialist care, although they are a frequent provider of concussion-related healthcare. Although we tried to reduce the likelihood of contamination of this cohort with individuals requiring pre-injury referrals to specialized physicians, the date the referral was made to the specialist was unknown, and, thus, some referrals may be unrelated to a concussion. This analysis also was unable to include injury-specific confounding variables such as duration and depth of LOC, post-traumatic amnesia (PTA), effects of medico-legal issues (e.g., workplace-related injuries or litigation from motor vehicle collisions), mechanism of injury, initial symptom presentation, or factitious disorders. The decision to use index years 2008 and 2014 for the Validation Cohort was made prior to initiation of the study and may have affected the performance of the validation model. Knowledge of concussions and their appropriate treatment in 2008 was considerably less than in subsequent years included in this study, and this may have influenced the validation.

## Conclusions

The results of the current study should allow the first primary care or emergency medicine providers who see an individual with concussion to quickly and easily determine a patient's risk of developing PPCS while still in the acute stage of injury. This, in turn, should result in earlier tailoring of treatment with concussion education and reassurance of high likelihood of a good outcome to low-risk individuals, while higher-risk patients might receive earlier psychological intervention, prescription for subthreshold aerobic exercise [41], and referrals to specialist physicians and allied healthcare professionals as needed.

## Supporting information

**S1 Checklist TRIPOD.**
(DOCX)

**S1 Table. Premorbid health conditions diagnostic codes.**
(DOC)

**S2 Table. Univariate logistic regressions in the Prolonged Concussion Symptom Cohort (2008–2014).**
(DOC)

**S1 Fig. Methodology for deriving risk factors for persisting post-concussion symptoms using the Ontario Concussion Cohort.** ED, emergency department; ENT, otolaryngology; NACRS; National Ambulatory Care Reporting System; OHIP, Ontario Health Insurance Plan; TBI, traumatic brain injury; TMJ, temporomandibular joint disorder.
(TIF)

**S2 Fig. Forward logistic regression modeling in Validation Cohort with selection stop step indicated (vertical line after bipolar disorder).** SBC, Schwarz Bayesian information Criterion

for model goodness of fit; lower SBC better fit.
(TIF)

**S1 Tool. TRICORDRR (Toronto Rehabilitation Institute Concussion Outcome Risk Determination and Rehab Recommendations) calculator tool.**
(DOC)

**S1 Text. Dataset Creation Plan (includes Analysis Plan).**
(PDF)

## Acknowledgments

This study was supported by ICES, which is funded by an annual grant from the Ontario Ministry of Health and Long-Term Care (MOHLTC). The authors also thank Charissa Levy, Refik Saskin, Dr. Jennifer Voth, Symron Bansal, Lisa Ellison, and Dr. Susan Jaglal for their assistance in compiling the cohort. Dr. Roger Zemek provided some literature. Dr. Varoon Thavapalan, Dr. Abe Snaiderman, and Sarah Cote provided assistance in editing.

The opinions, results and conclusions reported in this paper are those of the authors and are independent from the funding sources. No endorsement by ICES or the Ontario MOHLTC is intended or should be inferred. Parts of this material are based on data and information compiled and provided by Canadian Institute for Health Information (CIHI). However, the analyses, conclusions, opinions, and statements expressed herein are those of the author and not necessarily those of CIHI.

## Author Contributions

**Conceptualization:** Laura Kathleen Langer, David Wyndham Lawrence, Sarah Elizabeth Patricia Munce, Alice Kam, Alan Tam, Lesley Ruttan, Paul Comper, Mark Theodore Bayley.

**Formal analysis:** Laura Kathleen Langer, Seyed Mohammad Alavinia.

**Funding acquisition:** Mark Theodore Bayley.

**Investigation:** Laura Kathleen Langer, David Wyndham Lawrence, Alice Kam, Alan Tam, Lesley Ruttan, Paul Comper, Mark Theodore Bayley.

**Methodology:** Laura Kathleen Langer, Seyed Mohammad Alavinia, Sarah Elizabeth Patricia Munce, Mark Theodore Bayley.

**Project administration:** Laura Kathleen Langer.

**Validation:** Seyed Mohammad Alavinia.

**Visualization:** Laura Kathleen Langer, Mark Theodore Bayley.

**Writing – original draft:** Laura Kathleen Langer, Mark Theodore Bayley.

**Writing – review & editing:** Seyed Mohammad Alavinia, David Wyndham Lawrence, Sarah Elizabeth Patricia Munce, Alice Kam, Alan Tam, Lesley Ruttan, Paul Comper.

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
