## [Editor Report · Decision Letter 0]

10 Nov 2020

Dear Dr Langer, 

Thank you for submitting your manuscript entitled "Predicting Prolonged Post-Concussion Symptoms using the TRICORDRR (Toronto Rehabilitation Institute Concussion Outcome Risk Determination & Rehab Recommendations)" for consideration by PLOS Medicine.

Your manuscript has now been evaluated by the PLOS Medicine editorial staff as well as by an academic editor with relevant expertise, and I am writing to let you know that we would like to send your submission out for external peer review.

Kind regards,

Caitlin Moyer, Ph.D.,

Associate Editor

PLOS Medicine

---

## [Decision Letter · Decision Letter 1]

7 Jan 2021

Dear Dr. Langer,

Thank you very much for submitting your manuscript "Predicting Prolonged Post-Concussion Symptoms using the TRICORDRR (Toronto Rehabilitation Institute Concussion Outcome Determination and Rehab Recommendations)" (PMEDICINE-D-20-05392R1) for consideration at PLOS Medicine. 

Your paper was evaluated by a senior editor and discussed among all the editors here. It was also discussed with an academic editor with relevant expertise, and sent to three independent reviewers, including a statistical reviewer. The reviews are appended at the bottom of this email and any accompanying reviewer attachments can be seen via the link below:

[LINK]

In light of these reviews, I am afraid that we will not be able to accept the manuscript for publication in the journal in its current form, but we would like to consider a revised version that addresses the reviewers' and editors' comments. Obviously we cannot make any decision about publication until we have seen the revised manuscript and your response, and we plan to seek re-review by one or more of the reviewers. 

We expect to receive your revised manuscript by Jan 28 2021 11:59PM. Please email us (plosmedicine@plos.org) if you have any questions or concerns.

We look forward to receiving your revised manuscript. 

Sincerely,

Caitlin Moyer, Ph.D.

Associate Editor 

PLOS Medicine

plosmedicine.org

1. If possible, please compare the definition for PCS against more conventional criteria in a subset of randomly selected patients (e.g. a few hundred) where the latter are available as an embedded validation of the approach, or at least to show how it performs in comparison to more conventional approaches.

2.Title: Please revise your title according to PLOS Medicine's style. Your title must be nondeclarative and not a question. It should begin with the main concept if possible. "Effect of" should be used only if causality can be inferred, i.e., for an RCT. Please place the study design ("A randomized controlled trial," "A retrospective study," "A modelling study," etc.) in the subtitle (ie, after a colon).

3.Data Availability statement: Unfortunately, the link (www.ices.on.ca/DAS) did not seem to work when tried. For the ICES dataset, please provide a specific DOI/accession number needed to identify the relevant dataset, and if possible provide a contact email address (note this cannot be a study author) to request access. For the dataset creation plan and analytic code, please either include these as supporting information files, or provide a link where they can be accessed (the contact point for these cannot be one of the authors).

4.Abstract: Methods and Findings: Please start the first sentence of this section: “Data from a cohort study (Ontario Concussion Cohort study, 2008-2016; n=1,330,336) including all adults with a concussion diagnosis by either primary care physician… were used in a retrospective analysis” or similar.

5.Abstract: Methods and Findings: Please include some background demographic information (e.g. population and setting, ages, etc).

6.Abstract: Methods and Findings: At line 43: “The total cohort was divided into derivation…” Please also note the derivation and validation cohorts were not randomly split but divided based on year of index event.

7.Abstract: Methods and Findings: For the risk estimates for PPCS in the derivation cohort, please include the actual numbers underlying the percentages. At Line 49-50, please also describe the variables included in the models for the validation and derivation cohorts.

8.Abstract: Methods and Findings: In the last sentence of the Abstract Methods and Findings section, please describe the main limitation(s) of the study's methodology.

9.Abstract: Conclusions: We suggest beginning the paragraph with “In this study, we observed that premorbid psychiatric conditions…” or similar. 

10.Author Summary: Why was this study done?: Please revise the third point to: “The goal of the study was to create a risk calculation for every adult…”

11.Author Summary: What did the researchers do and find?: Please change 587.057 to 587,057

12.Author Summary: What do these findings mean?: We suggest combining the last two bullet points, and adjusting the language to avoid implying causality “Most adults with a concussion recover by 6 months following injury; however, our results suggest that pre-exisiting disorders are associated with increased likelihood of prolonged post-concussion symptoms.”

13. Methods: Ethics section: Please mention whether the requirement for informed consent was waived by the ethics board that approved the study.

14.Methods: Prospective analysis plan: Did your study have a prospective protocol or analysis plan? Please state this (either way) early in the Methods section.

15. Results: Please provide ORs, 95% CIs and p values for all relevant results presented in the main text. If adjusted analyses are presented, please also present the results from unadjusted models (these may be in a separate or supporting information table, and it should be noted in the text or figure legend when adjusted analyses are presented and which factors were included in the adjustment).

16. Discussion: Please revise the first sentence to begin: “To the best of our knowledge, this is the largest study to date…”

17. Discussion: Line 287: Please use square brackets for in text citations throughout.

18. Discussion: Line 324: Please adjust the reference within the bracket to [7,8].

19. Discussion: Line 349: Please clarify what is meant by the sentence: “This may be due to decreased neurocognitive reserve.”

20. Discussion: Line 374: Please clarify “effects of medico=legal issues”

21. References: Please use the "Vancouver" style for reference formatting, and see our website for other reference guidelines https://journals.plos.org/plosmedicine/s/submission-guidelines#loc-references

22. Table on Page 15: This appears to be the results of the Multivariate Logistic regression model in table format (from Figure 2). Please make this more clear (either incorporate into Figure 2 as a series of columns, or include a separate Table legend.

23. Table 1 on page 32: Please use consistent terms “Concussion Cohort” vs. Adult Cohort vs. Adult Concussion Cohort. Please define the abbreviation for GP, TBI, and TMD. Please indicate values that are N (%) vs. value (SD), for example.

24. As mentioned by reviewer 2, please do present the demographics for the derivation vs. validation cohort, either in the main table or a supporting information file.

24. Table 2: Please indicate “Primary Care” reflects numbers of visits.

25. Figure 2: Please provide an x axis label, and please provide a descriptive legend for the figure. Please define the abbreviation for TMJ.

26. Supplemental Figure 2: Please note in the legend that the stop step is indicated by the vertical line. Please define “SBC” and include a general X axis label for the figure.

27. Supporting information file TRICORDRR risk tool: Please spell out or define the abbreviation for OCD.

28. Supporting information file: Thank you for including the full text of your article “Increasing Incidence of Concussion: True Epidemic or Better Recognition?” however, it is not necessary to include as it is cited in the reference list.

29. Checklist: Please ensure that the study is reported according to the Transparent reporting of a multivariable prediction model for individual prognosis or diagnosis (TRIPOD) or the most appropriate reporting guideline for your study, and include the completed TRIPOD checklist as Supporting Information. Please add the following statement, or similar, to the Methods: "This study is reported as per the Transparent reporting of a multivariable prediction model for individual prognosis or diagnosis (TRIPOD) statement (S1 Checklist)."

The TRIPOD statement can be found here: https://www.equator-network.org/reporting-guidelines/tripod-statement/

Comments from the reviewers:

Reviewer #1: This is a very interesting and important paper that seeks to employ a large existing diagnostic health data system to determine factors associated with prolong post concussive symptoms. 

 The finings that behavioral health is associated with prolonged symptoms is not necessarily novel yet this paper expands this fining in a large population based setting. The limitations of the study are related to confirmation of diagnosis and unrecorded information as well as not using an a priori definition for post concussive symptoms.

1. It would be helpful for the authors to consider the fact that those with behavioral health co morbidities could be expressed as a series of somatic complaints and concerns. Might there be a population of people that had many of the ill defined PPCS symptoms preinjury or ascribed everything many of the concerns to the concussion that had existed previously

2. The authors may wish to add additional co morbid that could be linked to health complaints- OSA and overall BMI for example

3. The recent work of the TRACK TBO group may be a jumping off point for further discussion ie Nelson et al and Stein et al. ie the role of behavioral health and lack of resolution of symptoms in a prospectively collected cohort. In addition, Seabury et al speaks to the lack of ED follow up in a large group of those with brain injury. This paper creates an important point to consider is that the work in this paper may help to consider a " at risk cohort" that deserves clinical follow up early after injury

3a.The authors may to also construct a low risk predictor given the above Seabury et al data

Reviewer #2: "Predicting Prolonged Post-Concussion Symptoms using 1 the TRICORDRR (Toronto Rehabilitation Institute Concussion Outcome Determination and Rehab Recommendations)" describes the development and validation of a risk score to predict risk of needing prolonged concussion care. A cohort from the Ontario Concussion Cohort Study (2008-2016) was used, with the derivation cohort being from 2009-2013, and the validation cohort being from 2008 & 2014. Certain risk factors such as advanced age, bipolar disordor, high level of primary care visits per year, etc were identified. An AUC of 0.64 was achieved for the model applied to the validation cohort, and AUC=0.79 for the derivation cohort with bootstrap resampling.

The results suggest moderate efficacy of the risk score model towards concussion care prediction. Additionally, some issues might be addressed:

1. In Lines 50 to 52, the sentence "The area under the curve (AUC) for the derivation model was 0.79, Validation model was 0.64, and the bootstrap was 0.79" seems slightly awkward. It appears that the same final model was validated in all cases; if so it might be phrased more similarly to "the AUC the final model was 0.79 on the derivation dataset... validation dataset...", etc. Moreover, from line 197, it seems that the derivation cohort and bootstrap are actually one and the same. This might be clarified.

2. In Line 182, the details of the univariate logistic regressions etc. involved in variable selection, as well as the details of the bootstrap resampling procedure, might be presented - possibly in supplementary material. This is especially if the derivation may not follow exactly from Sullivan et al. (2004).

3. For the CART tree presented in Figure 4, the predictions/outcomes at each node of the tree might also be presented (as referred to in Line 196, with reference to Sullivan et al., 2004), as appears to be standard practice.

4. In line 204, "duplicate records" might be defined in more detail. If it refers to individuals with multiple diagnoses of concussion (e.g. first concussion in 2008, then in 2013), was there a particular reason for exclusion instead of including the first occasion? If not, the treatment for such patients with multiple diagnoses might be described further, especially as they are then possibly represented in both the derivation and validation dataset.

5. The choice of having the validation cohort being comprised of both diagnoses before (2008) and after (2014) the derivation cohort, might be explained further. While this seems that it would mitigate confounders (e.g. treatment conditions) across the study period to an extent, there remains a potential concern of validation on data prior to that used for derivation. As such, it might be considered to report the outcomes for 2014 separately from 2008 as well.

6. The demographics odds ratio table on Page 15 does not appear to be captioned.

7. For the corresponding demographics tables (Tables 1a & 1b), it might be considered to also present the figures separated according to derivation and validation cohorts, to determine as to whether they share similar distributions.

8. The derivation of the risk point system (Line 260, Table 2) might be described in greater detail, possibly in the supplementary material. If possible, the performance difference between a (more complicated logistic) model estimate based on raw odds ratios, and the approximated risk point system, might be presented.

9. In Line 268, Figure 6 is described as "the number of cases that had PPCS..."; might this be the proportion of cases instead?

10. The numerical details of the percentages/number of patients involved for each risk score strata in Figure 6 might be included in the supplementary material.

Reviewer #3: This manuscript reports the results of the study focused on analysis for predicting risk of prolonged postconcussion symptoms (PPCS). Specifically, the study uses the Toronto Rehabilitation Institute Concussion Outcome Determination and Rehab Recommendations (TRICORDRR) in their prediction model. Overall, the studies well done and likely to make an important contribution to the literature. The following comments are intended to help the authors improve the overall quality and impact of their manuscript.

1. The definition of PPC as used in this study, based on two or more specialist visits for concussion related symptoms more than six months after injury, is novel and unorthodox. Please provide justification for this definition.

2. It would be helpful to understand how accurate the premorbid health retrospective review was for this study, relying on relevant ICD-9 diagnostic codes. Obviously, premorbid conditions emerged as important predictor variables in the model, so the reliability of the original data is critical. Please comment

3. The greatest concern about this study has to do with the methodology used to calculate the rate of PPCS. That is, the calculation of the rate of prolonged postconcussion symptoms as 12.5% appears to be based on the denominator of only those patients that had two years of healthcare system tracking, not the original cohort of patients with concussion (n=1,330,336). My main concern is that this methodology artificially inflates the reported rate of PPCS. That is, many of the original concussed cohort patients may have had a spontaneous and complete recovery, which negated their need for any follow-up in the healthcare system after their concussion. If so, then the current estimated rate of 12.5% excludes them in his face only on the percentage of patients who pursue follow-up after their injury and continue to have prolonged symptoms at least six months after injury. The persisting cohort of 73,122 patients with persistent symptoms represents only 5.4% of the original concussed cohort of 1,330,336 patients. The different methodologies represent more than a twofold difference, 5.4% versus 12.5%. This issue is of major importance, as it has important bearing on the reported rate of PPCS from this study, which is likely to be cited frequently after publication. Please provide verification and justification for the methodology used. 

4. With respect to the results of your production model, please comment on how these findings from your very large study are novel or highly consistent with the existing literature. Specifically, comment on novel findings from this study after the existing literature on age and premorbid psychiatric factors predicting risk of prolonged postconcussion symptoms after mild traumatic brain injury.

5. The modeling statistics, results, and graphic illustration of the findings are of good quality.

[LINK]

---

## [Decision Letter · Decision Letter 2]

30 Mar 2021

Dear Dr. Langer,

Thank you very much for re-submitting your manuscript "The TRICORDRR (Toronto Rehabilitation Institute Concussion Outcome Determination and Rehab Recommendations): Targeting those at most risk for prolonged post-concussion treatment

Short Title: Risk Score for Prolonged Post-Concussion Symptoms" (PMEDICINE-D-20-05392R2) for review by PLOS Medicine.

I have discussed the paper with my colleagues and the academic editor and it was also seen again by two reviewers. Provided the remaining editorial and production issues are dealt with we are planning to accept the paper for publication in the journal.

[LINK]

We look forward to receiving the revised manuscript by .   

Sincerely,

Caitlin Moyer, Ph.D.

Associate Editor 

PLOS Medicine

plosmedicine.org

Requests from Editors:

1. Please fully address the remaining reviewer comments.

2. Title: Please revise your title according to PLOS Medicine's style. Your title must be nondeclarative and not a question. It should begin with main concept if possible. "Effect of" should be used only if causality can be inferred, i.e., for an RCT. Please place the study design ("A randomized controlled trial," "A retrospective study," "A modelling study," etc.) in the subtitle (ie, after a colon). We suggest Prediction of risk of prolonged post-concussion symptoms: Derivation and validation of the TRICORDRR (Toronto Rehabilitation Institute Concussion Outcome Determination and Rehab Recommendations) score” or something similar.

3. Data availability statement: Thank you for clarifying this, and please also include the reference number in the statement, alongside the ICES contact information (reference TRIM#201609760310000) if relevant for data requests.

4. Prospective analysis plan: Thank you for including the analysis plan as a supporting information file. Please reference the file in the Methods at Line 201 (S1_Analysis Plan). From looking at the analysis plan, it does not seem to describe in detail the specific analyses outlined in the study. In the Methods section, please fully describe when specific analyses were planned, and when/why any data-driven changes to analyses took place, including changes from peer review comments (these are described very generally at Lines 202-204).

5. Abstract: Line 41: Should this be “retrospective analysis” rather than retroactive?

6. Abstract: Please double check grammar/sentence structure throughout- for example: “The total cohort was divided…” at line 44, and “Variables were selected a priori... “ at line 46.

7. Abstract: Methods and Findings: Please clarify the values given in parentheses at lines 50-52: “Highest risk estimates for PPCS derived in the Derivation Cohort were: >61 years (0.54), bipolar disorder (0.52), high level of primary care visits per year (0.46), personality disorders (0.45), and anxiety and depression (0.33).”

8. Abstract: Line 54: Please define the abbreviation OHIP at first use.

9. Abstract: Conclusions: Please revise to avoid causal language: “In this study, we observed that premorbid psychiatric conditions were associated with increased risk of a prolonged recovery from concussion. Increased health care system usage and older age also were significantly associated with increased risk.”

10. Author summary: Please use sentence structure/punctuation for bullet points.

11. Author summary: We suggest revising under “What do these findings mean” to read: “The risk score may aid physicians treating adults with a concussion tp quickly assess the patient’s absolute risk of a good outcome or a prolonged recovery and to be able to tailor the treatment plan as appropriate, such as encouraging return to aerobic exercise, education about concussion, timely referrals for specialized psychological care, etc.” if relevant, to reflect the conclusion ties back to the derivation of the risk score.

12. Author summary: We suggest revising the final bullet point to avoid causal implications: “...psychiatric disorders and health care utilization are associated with increased risk of likelihood…”

13. Methods: Line 128-129: Can you please clarify this sentence: “Visible minorities accounted for 29.3% of the population.”

14. Results: Line 263-264: Please provide the p value for this, in addition to the 95% CIs, and please revise for causal language: The location of diagnosis (ED or primary care) was not significantly associated with the number or type of specialist visits for persisting symptoms (OR 1.00, CI 0.996 – 1.006).

15. Results: Line 310: Please clarify to “...hypothetical patient aged 50-60 years…”

16. Results: Line 324: If it has not been previously spelled out in the text to this point, please fully define the abbreviation for TRICORDRR here at first use.

17. Discussion: We suggest opening with a discussion of the main findings of the study.

18. Discussion: Line 394: Please include a space between “aging” and the reference bracket. (Please also correct at line 402, and double check throughout).

19. Discussion: Line 419-420: Please revise to “...it has the highest number of visits with specialist physicians compared with other premorbid conditions…” Please also elaborate on the “decreased neurocognitive reserve” here.

20. Discussion: Line 434: Please capitalize Black in this sentence.

21. Discussion: Future Directions: Please clarify that the TRICORDRR is “the tool” indicated, and also it would be helpful to include a sentence putting into context what these ongoing studies hope to inform/accomplish.

22. References: Please double check all formatting. Please use the "Vancouver" style for reference formatting, and see our website for other reference guidelines https://journals.plos.org/plosmedicine/s/submission-guidelines#loc-references

For example, please check journal name and punctuation in 11, 12, 13 (throughout).

23. Supplemental Figure 1: Please define all abbreviations used in the legend. If possible, please provide a short description in the legend to help illustrate the main point.

24. TRIPOD checklist: Please include the locations in the text where each item can be found, rather than checking the points off the list. Please use section and paragraphs, rather than page numbers, to refer to locations. Please include a separate supporting information file Title/label for this (e.g. S1_Checklist).

25. Figure 2: In the X axis label, please change to “Odds Ratio”

26. Figure 3: Please provide a descriptive legend for the figure, making it slightly more clear what point is illustrated.

27. Figure 4: Please provide a descriptive legend for this figure.

28. Figure 5: Please provide a descriptive legend for the figure, making it slightly more clear what point is illustrated.

Comments from Reviewers:

Reviewer #2: We thank the authors for considering our previous comments. However, some concerns remain:

1. Given that the motivation for selecting 2008 and 2014 for validation was specifically that "...they were expected to be the best and worst years and did not want it to influence the derivation cohort", it could be appropriate to (also) present their analyses separately, possibly in supplementary material. This is since neither were expected to represent typical performance, and it is uncertain as to whether their combination into a single validation set would represent the typical cohort either. It might be noted that this choice could have contributed to the reduced validation performance of the model, which might be considered as a limitation.

Minor issues:

2. The sentence beginning in Line 52 might be "The area under the curve (AUC) was 0.79 for the derivation model, 0.79 for bootstrap internal validation of the Derivation Cohort, and 0.64 for the Validation model."

3. For the CART figure, the explanation pertaining to SAS graphical limitations are acknowledged, but it would remain recommended to extract the figures corresponding to the nodes if at all possible (e.g. by writing the values of the corresponding variables to the output log)

Reviewer #3: I appreciate the authors' response to reviewer comments and their respective manuscript revisions. These modifications have significantly improved the overall quality and impact of the manuscript.

[LINK]

---

## [Editor Report · Decision Letter 3]

7 May 2021

Dear Dr Langer, 

On behalf of my colleagues and the Academic Editor, David Menon, I am pleased to inform you that we have agreed to publish your manuscript "Prediction of risk of prolonged post-concussion symptoms: Derivation and validation of the TRICORDRR (Toronto Rehabilitation Institute Concussion Outcome Determination and Rehab Recommendations) score" (PMEDICINE-D-20-05392R3) in PLOS Medicine.

Also, please address the following editorial points:

-Author summary: Line 81: Please revise "concussed adults" to "adults with concussion" or similar.

-Methods: Line 131: Please avoid the use of the Statistics Canada quotation if possible. If necessary to use, please include a reference. We suggest avoiding the term "Caucasian" and instead use "...persons, other than Aboriginal peoples, who are non-White..." or similar.

-TRIPOD Checklist: Please revise the Checklist. Please do not use page or line numbers to refer to locations in the text. Please use sections and paragraph numbers to refer to locations in the text for each item (for example "Methods, paragraph 2").

-Reference list: Please check formatting of references 21, 22, 27, 30, 31, 35, 41 for correct journal abbreviations and punctuation.

PRESS

Sincerely, 

Caitlin Moyer, Ph.D. 

Associate Editor 

PLOS Medicine